# Impacts of a sugar sweetened beverage tax on body mass index and obesity in Thailand: A modelling study

**Payao Phonsuk**[1]ⓘ*, **Vuthiphan Vongmongkol**[1‡], **Suladda Ponguttha**[1‡],
**Rapeepong Suphanchaimat**[1,2]ⓘ, **Nipa Rojroongwasinkul**[3‡], **Boyd Anthony Swinburn**[4]ⓘ

**1** International Health Policy Program, Nonthaburi, Thailand, **2** Department of Disease Control, Ministry of Public Health, Nonthaburi, Thailand, **3** Institute of Nutrition, Mahidol University, Nakhon Pathom, Thailand, **4** School of Population Health, University of Auckland, Auckland, New Zealand

ⓘ These authors contributed equally to this work.
‡ These authors also contributed equally to this work.
* payao@ihpp.thaigov.net

**Data Availability Statement:** Regarding the data availability of the data source we used in this study,

## Abstract

### Background

The World Health Organization (WHO) recommends sugar-sweetened beverage (SSB) taxes to address obesity. Thailand has just launched the new tax rates for SSB in 2017; however, the existing tax rate is not as high as the 20% recommended by the WHO. The objective for this study was to estimate the impacts of an SSB tax on body mass index (BMI) and obesity prevalence in Thailand under three different scenarios based on existing SSB and recommended tax rates.

### Methods

A base model was built to estimate the impacts of an SSB tax on SSB consumption, energy intake, BMI, and obesity prevalence. Literature review was conducted to estimate pass on rate, price elasticity, energy compensation, and energy balance to weight change. Different tax rates (11%, 20% and 25%) were used in the model. The model assumed no substitution effects, model values were based on international data since there was no empirical Thai data available. Differential effects by income groups were not estimated.

### Findings

When applying 11%, 20%, and 25% tax rates together with 100% pass on rate and an -1.30 own-price elasticity, the SSB consumption decreased by 14%, 26%, and 32%, respectively. The 20% and 25% price increase in SSB price tended to reduce higher energy intake, weight status and BMI, when compared with an 11% increase in existing price increase of SSB. The percentage changes of obesity prevalence of 11%, 20% and 25% SSB tax rates were estimated to be 1.73%, 3.83%, and 4.91%, respectively.

this study is based on the secondary data analysis. The SSB consumption and weight and height data among Thai is derived from the food consumption data survey in 2016 which fully belongs to the National Bureau of Agricultural Commodity and Food Standard, Thailand (ACFS). The ACFS has the report on its survey results publicly available. However, the raw data are set are upon the request. In our protocol, we have sent the letter to the ACFS office and have requested the data and variables we need for the analysis. The ACFS allows using the data only in this study. We could only provide the minimal set of baseline data for SSB consumption and BMI, by sex and age groups in the supporting information (S1 Table and S2 Table). In order to get the raw data of the survey, please kindly contact the ACFS at telephone 0066-2561-2277 or e-mail address at itc@acfs.go.th.

**Funding:** This study was funded by the Food and Nutrition Policy for Health Promotion Program (FHP), ThaiHealth Promotion Foundation, and the Thailand Science Research and Innovation (TSRI) under the Senior Research Scholar on Health Policy and System Research [Contract No. RTA6280007]. The funders have no role in study design, data collection and analysis, decision to publish, or preparation of the manuscript.

**Competing interests:** The authors have declared that no competing interests exist.

**Abbreviations:** BMI, body mass index; SSB, sugar-sweetened beverage.

## Conclusions

A higher SSB tax (20% and 25%) was estimated to reduce consumption and consequently decrease obesity prevalence. Since Thailand has already endorsed the excise tax structure, the new excise tax structure for SSB should be scaled up to a 20% or 25% tax rate if the SSB consumption change does not meet a favourable goal.

## Introduction

Obesity is increasing in all countries including Thailand. The prevalence of obesity among Thai adults has increased significantly from 28% to 33% in men, and from 41% to 42% in women, between 2008 and 2014 [1]. A similar obesity trend can also be observed among Thai children under 5 years where the prevalence increased from 6.9% in 2005 [2] to 8.2% in 2015 [3].

A high intake of sugar-sweetened beverages (SSB) is considered an important risk factor for obesity [4], diabetes, and cardiovascular diseases [5,6]. A cohort study in Thailand confirmed that consuming SSB at least once a day could result in gaining weight by 0.5 kg [7]. Daily SSB consumption in Thailand continues to increase among children, from 8.7% in 2003 to 17.2% in 2008–2009, and adults, from 5.1% to 7.9% in the corresponding period [8]. Additionally, data from the Euromonitor also showed an increase in SSB sales volume between 2005 and 2015. In 2015, the sales volume was as large as 4,100 million litres, and it was forecasted that the sales of SSB would be increased to 22% by 2020 [9].

The World Health Organization (WHO) recommended SSB taxation as one of the 'good buy' interventions to prevent overweight and obesity. According to the recommendations, countries should aim to escalate the retail price of SSB by at least 20% using an imposed taxation [10]. Many countries have either implemented an SSB tax or been showing interest in doing so [11–13]. Evidently, in Mexico, 1 peso taxation per litre of SSB (equivalent to a 10% increase in total price) resulted in reduced SSB purchasing by 12% [14]. Evidence also suggested that an SSB tax correlated with a decline in SSB purchase and consumption, from 10% to 45% [15–20]. Moreover, numerous studies also extrapolated that overweight and obesity prevalence would be reduced ranging from 1% to 5% if applying a 20% tax [11,13,15,19,21,22].

Thailand has implemented an excise tax on non-alcoholic beverages since 1984. The excise rates were based on both ad valorem (valued based) rate and a specific volume-based rate where greatest payable revenue was selected. It was noted that the sugar-containing beverages were taxed at a lower degree in comparison with non-sugar beverages. Subsequently, the Excise Act BE 2560 announced a reformulation of the excise tax for SSB in September, 2017 in order to reduce sugar consumption and improve the health of populations. A tiered tax approach was then introduced where both ad valorem and specific tax on sugar content were used for the tax calculation. A significant change in the new tax regulation is that the ad valorem rate was reduced from 20% to a range of 0–14%, based on the type of beverages (0% for beverage concentrates, 10% for fruit and vegetable juice and 14% for soda and carbonated drinks). Additionally, specific tax rates were adjusted to be based on sugar content, with a ranging from less than 6 g to the highest amount of more than 18 g. An SSB with more than 6 g per 100 ml will be levied by a higher tax rate than those with lower sugar concentration. For example, a carbonated drink with 6 g per 100 ml of added sugar will be ad valorem taxed by 14%, with no a specific taxing, whereas a same product with 10 g of sugar per 100 ml will be charged with a 14% ad valorem tax and 0.30 Baht per litre for specific tax. This system also

stated the specific tax rates would increase after 2019 and every two years afterward and settle in 2023 with a maximum rate of 5 Baht per litre for fruit and vegetable juice, soda, and carbonated drinks and 44 Baht per litre for beverage concentrates [23]. As a result, the tax rates for each type of beverage can vary according to the Act. However, a grace period is provided for beverage industries to gradually reduce sugar content to meet the tax threshold. Since 1 October, 2019 till present, Thailand has been in the second phase of new tax-policy implementation.

Recent study investigated an impact of the new tax rates showed that the SSB prices were increased by 11% and sugar content among taxed SSB from both domestic and imported products were decreased by 10% [24]. So far, there has not been much research on the health impacts of an SSB tax, especially in low- and middle- income countries including Thailand. Thus, the aim of this study was to estimate the impacts of an SSB tax on body mass index (BMI) and obesity prevalence in Thailand under the different scenarios of various tax rates.

## Methods

An economic-epidemiologic mathematical model, using secondary data from the previous cross-sectional national survey by the National Bureau of Agricultural Commodity and Food Standard, Thailand (ACFS) in 2016, was applied to estimate the impact of an SSB tax on BMI and obesity prevalence. The model was based on the causal pathway framework which has been used in many countries, such as Ireland, the U.K., South Africa, and the U.S. [12,13,15,25] (Fig 1). Recent price change of 11% in SSB products in Thailand was used [24].

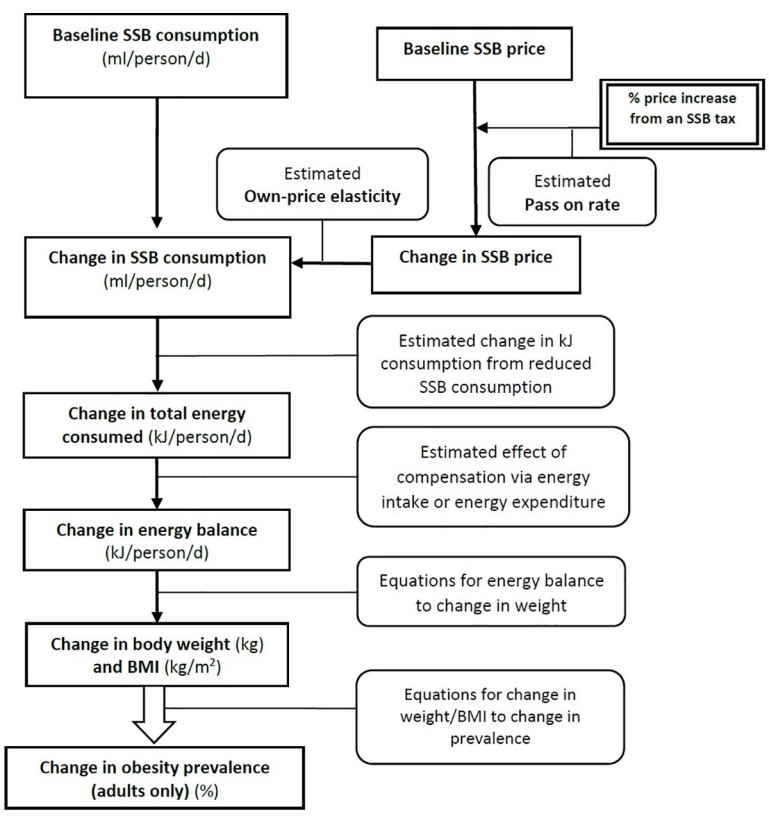

**Fig 1. Analysis framework.**

The 20% and 25% increase in price of SSB was selected based on the WHO recommendations [10].

## Operated definitions and data sources

The main parameters used in the model were pass on rate, price elasticity, baseline SSB consumption, and baseline BMI.

a. **Pass on rate.** This indicates the percentage of tax margin that the food chain passes on to the consumers. It ranges from less than, equal to, or greater than 100%. In other words, tax can be over- or under-shifted depending on the manufacturers, retailers and supply and demand chains. Empirical studies provided a variety of pass on rates when SSB taxes were implemented. For example, evidence from France showed a full pass on rate of the tax in soda prices, while flavoured water and fruit drinks demonstrated 85% and 60% pass on rate, respectively [26]. A study from Mexico suggested that the SSB tax in all types of beverage was fully passed on to consumers [27]. Since there is no prior research identifying an appropriate pass on rate in Thailand, this study therefore assumed that the pass on rate was 100%.

b. **Price elasticity.** Price elasticity is a parameter for estimating the change of SSB purchasing and consuming of when the price is increased from the taxation. Own-price elasticity refers to the change in purchasing if the price of the same product is changed. Cross-price elasticity is used to define the change in purchasing when a price of another product is altered [28]. With limited empirical data on price elasticity of SSB in Thailand, this study employed the figure of own-price elasticity from a recent meta-analysis on the impacts of SSB taxes on consumption and obesity prevalence by Cabrera- Escobar et al., where the data were collected from various countries including middle-income countries like Brazil and Mexico [29], where the economic context was similar to Thailand. The meta-analysis suggested an own-price elasticity of SSB at -1.30 (95% CI = [-1.089 to -1.509]), meaning that a 10% increase in price would decrease consumption by 13%. Note that this study did not include cross-price elasticity in the analysis due to data unavailability in Thailand.

c. **Baseline SSB consumption.** The data from the ACFS cross-sectional survey in 2016 was used to estimate the baseline SSB consumption among Thais. The survey employed stratified three-stage sampling with a sample size of 6,998. Food consumption was estimated using a semi-quantitative food frequency questionnaire. SSB was defined as a non-alcoholic beverage with added sugar, including a) carbonated soft drinks, b) sport drinks, c) energy drinks, d) ready-to-drink (RTD) tea and coffee, and e) sweet drinks and fruit juice. Participants were asked to report the frequency and the portion size of the drinks they consumed during the last month. Then, the frequency was multiplied by the portion size to quantify SSB daily intake. The sugar contents of SSB products were calculated based on food composition data from the Food and Drug Administration (FDA), Thailand. The average sugar content of all SSB was 12 g per 100 ml [30]. The energy content in kilojoules (kJ) and kilocalories (kcal) was calculated based on grams of sugar content.

d. **Baseline BMI.** Data on height and weight from the same samples from the ACFS survey in 2016 were used to estimate weight status and mean BMI among Thais. Height and weight of the samples in the survey were objectively measured. The data on height and weight were used to calculate the average BMI ($kg/m^2$) for each age group.

## Modelling techniques

Based on the framework, the pass on rate was used to determine the change in price of SSB. Together with the estimation of price elasticity, the change in SSB consumption in three different taxes scenarios were presented. The change in total calories consumed was then derived, and it was used to estimate the change in body weight, based on the equations for children and adults. Consequently, the changes in BMI and obesity prevalence were calculated.

The model was used to compare two populations between the baseline population as reference (no SSB tax imposed) and the population exposed to a 11%, 20%, and 25% SSB tax rate. A Multivariate Analysis of Variance (MANOVA) technique was applied to test the differences of consumption, weight and BMI across tax rates. STATA version 14.2 was used to run the model. Details for each analysis step are presented as follows;

1. **Change in SSB consumption and energy intake.** A 11%, 20% and 25% price increase with a 100% pass on rate and own-price elasticity of -1.30 were used to estimate the change in SSB consumption. The SSB consumption was measured under a unit of ml per person per day, and was calculated for each sex and age group. This information was then used to estimate a change in sugar consumption and later the energy intake of SSB. It was noted that energy intake presented here referred only to the energy intake from SSB consumption. Compensatory changes in energy intake from other food and beverage sources or energy expenditure from physical activity were not included in the model [31].

2. **Change in weight status and BMI.** The change in weight for young children aged between 3 and 5 years was calculated based on the coefficients for change in weight per change in energy intake according to Long et al. [25]. These coefficients were originally proposed for calculating the change in basal metabolic rate [32] and physical activity levels [33] among children aged 2–4 years in prior research. The coefficients were 216 kJ/day/kg for boys and 204 kJ/day/kg for girls [25]. For children and adolescents aged 6–17 years, the equations from Hall et al. [34] were used. Hall and colleagues developed age- and gender-specific linear equations to predict the weight gain from a given energy imbalance. For children aged 7–17 years, the equations were kcal/day/kg = $68 - 2.5^* age$ for males and kcal/day/kg = $62 - 2.2^* age$ for females. For adults aged 18 years and older, the change in weight status and BMI was estimated using the equation from Hall et al. which suggested that changes in energy intake of 100 kJ per day lead to approximately 1 kg of weight change. This rule was applied for both men and women. It was noted that this change in weight would take a year for a 50% achievement and three years for a 95% of total weight changes [35]. However, this study applied a counterfactual which assumed only two different steady states (baseline population as the reference and the population as exposed to SSB tax). It was assumed that the weight change would reach 95% of an estimation.

3. **Change in obesity prevalence.** The mean BMI among all adults aged 18 years and older in both men and women was used to estimate obesity prevalence (BMI $\geq$ 25 kg/m$^2$) [36]. The baseline obesity prevalence was compared with the obesity prevalence at 11%, 20%, and 25% tax rates. Note that this study did not estimate obesity prevalence for children due to the absence of standard equations.

## Sensitivity analysis and scenarios of interest

There were nine scenarios of interest, including base model for a 20% tax rate and eight scenarios from sensitivity analysis. A sensitivity analysis was undertaken upon two of the main assumptions which accounted for the most uncertainty in the best estimate. The first

**Table 1. Sensitivity analysis scenarios by changing pass on rate and price elasticity.**

| Price elasticity | Pass on rate | | |
|---|---|---|---|
| | 50% | 100% | 150% |
| -0.94 | Scenario A | Scenario B | Scenario C |
| -1.30 | Scenario D | Base model for a 20% tax rate | Scenario E |
| -2.25 | Scenario F | Scenario G | Scenario H |

assumption was that the assumed 100% pass on rate of SSB tax was changed to 50% and 150%, representing under-shifting and over-shifting. The second assumption was the price elasticity which was initially assumed to be -1.30 was changed to -0.94, based on the studies from India [11] and -2.25, based on a modelling study in the U.S. [17]. Note that the tax rate used in the sensitivity analysis was 20%. Table 1 shows the eight additional scenarios of sensitivity analysis after changing the pass on rate and price elasticity.

## Results

### Change in SSB consumption and energy intake

Based on the framework, when applying 11%, 20%, and 25% increase in SSB price together with a 100% pass on rate and an own-price elasticity of -1.30, the SSB consumption decreased by 14%, 26%, and 32%, respectively. Table 2 shows the estimated mean reduction in energy intake from a 11%, 20%, and 25% SSB tax, classified by sex and age groups. A 25% increase in SSB price showed the highest reduction in energy intake. Men had a larger mean reduction in energy intake when compared with women in all tax scenarios. Young children and elderly had a smaller degree of reduction in energy intake than other age groups in both sexes.

### Changes in weight and BMI

The negative change in energy intake contributed to a reduction in weight and BMI (Tables 3 and 4). Overall, a 11%, 20%, and 25% increase in SSB price resulted in reductions in weight by

**Table 2. Estimated change in energy intake (kJ/day/person) after a 11%, 20% and 25% SSB tax by sex and age groups.**

| Sex | Age groups (years) | Tax 11% Mean (95% CI) | Tax 20% Mean (95% CI) | Tax 25% Mean (95% CI) |
|---|---|---|---|---|
| **Males** | 3–5 | -42.6 (-47.2, -38.0) | -79.1 (-87.6, -70.5) | -97.3 (-107.8, -86.8) |
| | 6–12 | -64.9 (-70.0, 59.8) | -120.5 (-130.0, -111.0) | -148.3 (-160.0, -136.6) |
| | 13–17 | -108.1 (-116.7, -99.4) | -200.7 (-216.7, -184.7) | -247.0 (-266.7, -227.3) |
| | 18–34 | -120.7 (-130.7, -110.6) | -224.1 (-242.8, -205.4) | -275.8 (-298.8, -252.8) |
| | 35–64 | -56.4 (-62.6, -50.2) | -104.8 (-116.2, -93.3) | -129.0 (-143.1, -114.8) |
| | 65 or older | -20.8 (-24.2, -17.4) | -38.6 (-44.9, -32.3) | -47.5 (-55.3, -39.7) |
| | **All men** | **-67.0 (-69.9, -64.1)** | **-124.4 (-129.8, -119.1)** | **-153.2 (-159.8, -146.5)** |
| **Females** | 3–5 | -38.1 (-42.3, -33.9) | -70.8 (-78.6, -63.0) | -87.1 (-96.7, -77.5) |
| | 6–12 | -63.5 (-69.6, -57.3) | -117.9 (-129.2, -106.5) | -145.0 (-159.0, -131.1) |
| | 13–17 | -83.7 (-90.8, -76.7) | -155.5 (-168.5, -142.5) | -191.4 (-207.4, -175.4) |
| | 18–34 | -83.3 (-90.7, -76.0) | -154.8 (-168.4, -141.1) | -190.5 (-207.3, -173.7) |
| | 35–64 | -35.1 (-39.5, -30.7) | -65.2 (-73.4, -57.1) | -80.3 (-90.3, -70.3) |
| | 65 or older | -10.0 (-11.9, -8.2) | -18.6 (-22.0, -15.2) | -22.9 (-27.1, -18.7) |
| | **All women** | **-51.5 (-53.9, -49.2)** | **-95.7 (-100.2, -91.3)** | **-117.8 (-123.3, -112.4)** |
| | **Total** | **-59.0 (-60.9, -57.2)** | **-109.6 (-113.1, -106.2)** | **-134.9 (-139.2, -130.7)** |

**Note:** 95% CI = 95% Confidence Interval.

**Table 3. Estimated change in weight (kg) after a 11%, 20% and 25% SSB tax by sex and age groups.**

| Sex | Age groups (years) | Tax 11% Mean (95% CI) | Tax 20% Mean (95% CI) | Tax 25% Mean (95% CI) |
|---|---|---|---|---|
| **Males** | 3–5 | -0.20 (-0.22, -0.18) | -0.37 (-0.41, -0.33) | -0.45 (-0.50, -0.40) |
| | 6–12 | -0.36 (-0.39, -0.33) | -0.67 (-0.72, -0.61) | -0.82 (-0.89, -0.75) |
| | 13–17 | -0.90 (-0.98, -0.83) | -1.68 (-1.82, -1.54) | -2.07 (-2.23, -1.90) |
| | 18–34 | -1.21 (-1.31, -1.11) | -2.24 (-2.43, -2.05) | -2.76 (-2.99, -2.53) |
| | 35–64 | -0.56 (-0.63, -0.50) | -1.05 (-1.16, -0.93) | -1.29 (-1.43, -1.15) |
| | 65 or older | -0.21 (-0.24, -0.17) | -0.39 (-0.45, -0.32) | -0.47 (-0.55, -0.40) |
| | **All men** | **-0.55 (-0.58, -0.52)** | **-1.02 (-1.07, -0.97)** | **-1.26 (-1.32, -1.20)** |
| **Females** | 3–5 | -0.19 (-0.21, -0.17) | -0.35 (-0.39, -0.31) | -0.43 (-0.47, -0.38) |
| | 6–12 | -0.38 (-0.21, -0.17) | -0.71 (-0.78, -0.64) | -0.87 (-0.96, -0.78) |
| | 13–17 | -0.70 (-0.76, -0.64) | -1.30 (-1.41, -1.19) | -1.60 (-1.74, -1.47) |
| | 18–34 | -0.83 (-0.91, -0.76) | -1.55 (-1.68, -1.41) | -1.90 (-2.07, -1.74) |
| | 35–64 | -0.35 (-0.40, -0.31) | -0.65 (-0.73, -0.57) | -0.80 (-0.90, -0.70) |
| | 65 or older | -0.10 (-0.12, -0.08) | -0.19 (-0.22, -0.15) | -0.23 (-0.27, -0.19) |
| | **All women** | **-0.42 (-0.44, -0.40)** | **-0.78 (-0.82, -0.74)** | **-0.96 (-1.01, -0.92)** |
| | **Total** | **-0.48 (-0.50, -0.47)** | **-0.90 (-0.93, -0.87)** | **-1.11 (-1.15, -1.07)** |

**Note:** 95% CI = 95% Confidence Interval.

0.48 kg (95% CI = [-0.50 kg, -0.47 kg]), 0.90 kg (95% CI = [-0.93 kg, -0.87 kg]), and 1.11 kg (95% CI = [-1.15 kg, -1.07 kg]), respectively. For the tax rate of 11%, the estimated mean change in weight was -0.55 kg (95% CI = [-0.58 kg, -0.52 kg]) among men and -0.42 kg (95% CI = [-0.44 kg, -0.40 kg]) among women.

The estimated change in mean BMI was also decreased in three different scenarios (Table 4). In general, the negative change in BMI in men was larger than women. By applying a 11% of SSB tax, BMI was decreased by 0.23 kg/m$^2$ (95% CI = [-0.24 kg/m$^2$, -0.22 kg/m$^2$]) and 0.20 kg/m$^2$ (95% CI = [-0.21 kg/m$^2$, -0.19 kg/m$^2$]) for men and women, respectively.

**Table 4. Estimated change in BMI (kg/m2) after a 11%, 20% and 25% SSB tax by sex and age groups.**

| Sex | Age groups (years) | Tax 11% Mean (95% CI) | Tax 20% Mean (95% CI) | Tax 25% Mean (95% CI) |
|---|---|---|---|---|
| **Males** | 3–5 | -0.18 (-0.20, -0.16) | -0.33 (-0.37, -0.30) | -0.41 (-0.45, -0.36) |
| | 6–12 | -0.20 (-0.21, -0.18) | -0.37 (-0.39, -0.34) | -0.45 (-0.49, -0.41) |
| | 13–17 | -0.33 (-0.36, -0.31) | -0.62 (-0.67, -0.57) | -0.76 (-0.82, -0.70) |
| | 18–34 | -0.42 (-0.46, -0.39) | -0.78 (-0.85, -0.72) | -0.96 (-1.04, -0.88) |
| | 35–64 | -0.21 (-0.23, -0.18) | -0.38 (-0.42, -0.34) | -0.47 (-0.52, -0.42) |
| | 65 or older | -0.08 (-0.09, -0.07) | -0.15 (-0.17, -0.12) | -0.18 (-0.21, -0.15) |
| | **All men** | **-0.23 (-0.24, -0.22)** | **-0.43 (-0.45, -0.41)** | **-0.53 (-0.55, -0.50)** |
| **Females** | 3–5 | -0.17 (-0.19, -0.15) | -0.32 (-0.35, -0.28) | -0.39 (-0.43, -0.35) |
| | 6–12 | -0.20 (-0.22, -0.18) | -0.38 (-0.41, -0.34) | -0.46 (-0.51, -0.42) |
| | 13–17 | -0.29 (-0.31, -0.26) | -0.53 (-0.58, -0.49) | -0.65 (-0.71, -0.60) |
| | 18–34 | -0.34 (-0.37, -0.31) | -0.63 (-0.68, -0.57) | -0.77 (-0.84, -0.70) |
| | 35–64 | -0.15 (-0.17, -0.13) | -0.28 (-0.31, -0.24) | -0.34 (-0.38, -0.30) |
| | 65 or older | -0.04 (-0.05, -0.04) | -0.08 (-0.10, -0.07) | -0.10 (-0.12, -0.08) |
| | **All women** | **-0.20 (-0.21, -0.19)** | **-0.36 (-0.38, -0.35)** | **-0.45 (-0.47, -0.43)** |
| | **Total** | **-0.21 (-0.22, -0.21)** | **-0.40 (-0.41, -0.38)** | **-0.49 (-0.50, -0.47)** |

**Note:** 95% CI = 95% Confidence Interval.

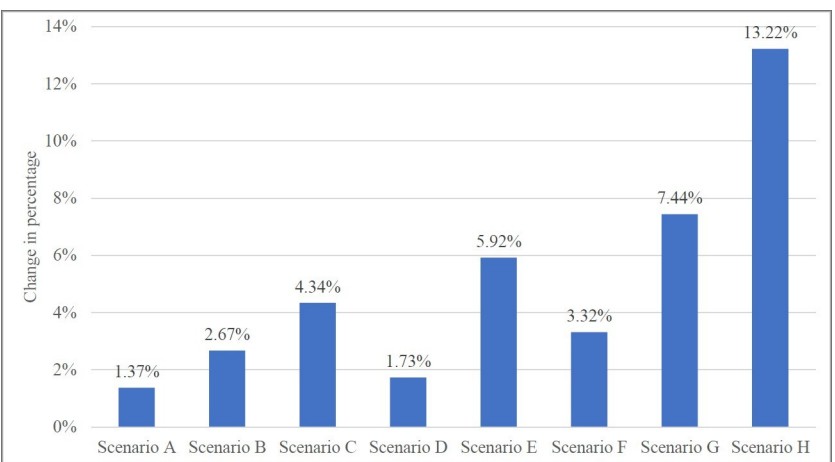

**Fig 2. Percentage change of obesity prevalence (BMI $\geq$ 25 kg/m$^2$) from eight sensitivity analysis scenarios.**

Furthermore, MONOVA analysis showed a significant difference (P-value<0.0001) in weight and change in BMI in all three tax-based scenarios across age groups and sex (S3 Table).

### Change in obesity prevalence

The estimated mean BMI from the baseline and mean BMI changes caused by a 11%, 20% and 25% SSB tax were predicted for people aged 18 years or above. The percentage changes of obesity prevalence of a 11%, 20% and 25% SSB tax were estimated at 1.73% (95% CI = [1.11%, 2.57%]), 3.83% (95% CI = [2.88%, 4.98%]), and 4.91% (95% CI = [3.83%, 6.18%]), respectively. The percentage change in obesity prevalence was higher among women than men.

### Sensitivity analysis and scenarios of interest

A 20% tax rate was chosen to run the sensitivity analysis. Overall, the models with a lower pass on rate and price elasticity yielded lower degree of changes in SSB consumption, energy intake, weight status and BMI in all populations (Fig 2). The inverse trend was observed when a larger pass on rate and price elasticity were applied. In short, higher pass on rate together with higher price elasticity presented a greater impact on consumption behaviour and health outcome. The sensitivity analysis thus confirmed that the changes in SSB consumption, energy intake, bodyweight, and obesity prevalence were significantly sensitive to both pass on rate and price elasticity.

## Discussion

Higher SSB tax rates resulted in a better estimated reduction in calories intake, weight and BMI, and greater obesity prevalence reduction. A recent 11% tax rate resulted in smaller weight change in comparison with the recommending 20% and 25% tax rates (0.48 kg, 0.90 kg, and 1.11 kg, respectively). A change in BMI also yielded a similar trend as it was forecasted that 11%, 20% and 25% tax rates would contribute to percentage decline in the obesity prevalence by 1.73%, 3.84% and 4.91%, respectively.

These findings from the Thai model are in the same ballpark as other studies as the similar parameters (such as tax rate, pass on rate, and price elasticity) were used [12,13,15,19]. However, the amount of energy reduction was rather unique across the studies due to the variation in baseline SSB consumptions in different populations. For example, American teenagers aged

15–19 years consumed about 1,100 kJ per day of SSB at the baseline level [25], whereas SSB consumption among Thai teenagers aged 13–17 years was approximately less than 800 kJ per day [37].

Another potential reason for the variation in energy intake might be the definition of SSB, which was diverse across all studies. This study defined SSB as carbonated soft drinks, sport drinks, energy drinks, ready-to-drink tea and coffee, and sweet drinks and fruit juice that contain less than 50% fruit juice concentration. However, in some studies, SSB referred to only carbonated soft drinks and fruit juices [12,13]. As a result, it could potentially affect the baseline energy intake in total volume among the studies, and later result in a higher energy intake for the study that included a wider variety of SSB products. Furthermore, the baseline energy intake in this study was derived from the food composition data from Thai FDA, which might be a different data source characteristic from other countries.

The reductions in energy intake and weight status led to a reduction in BMI. The decreased trend of BMI change observed in the Thai model was consistent with various literature from other countries [12,13,19,25,38–40], which different degrees of reduction has been found. For example, a study from the U.K. estimated a 0.07 BMI point decrease across the UK population after introducing a 20% SSB tax [15]. The modelling study from Mexico suggested the BMI reduction of 0.31 kg/m$^2$ as an impact of a 20% SSB tax [40]. The subtle differences between this study and those from other countries might be explained by the differences in baseline SSB consumption as mentioned earlier and differences in base BMI across populations.

## Strengths of the study

This study is one of a few studies that assessed an impact of SSB tax on population health in low- and middle-income countries and also in Thailand that have assessed the impact of SSB tax on population health. The model used in this study is recent and widely accepted internationally [15,25,41,42]. It used the best available evidence to determine consumption patterns and a validated set of equations to calculate the effects of a change in energy intake on body weight, BMI, and obesity prevalence. The baseline SSB consumption data from the ACFS survey was derived from the semi-quantitative food frequency questionnaire (Semi-FFQ) method in which the quantity of food and the frequency of consumption were identified [43]. Finally, this study analysed the differential effects of three different tax rates across sex and age groups.

## Limitations

Firstly, this study did not account for possible substitution effects, in other words, cross-price elasticity was not included in the model. In reality, people may change their SSB consumption behaviours in response to the change in SSB price. There was also a possibility that caloric food or beverages (such as fruit juice, milk, or ice cream) could be consumed as a substitution for SSB. For example, Fletcher et al. found that consumers might respond to the addition in SSB tax by increasing consumption of other foods and drinks to compensate for the reduced calories from the tax-induced SSB products [44]. Conversely, some argued that the SSB tax would not lead to substitution of other foods or beverages. Finkelstein et al. observed no relationship between the SSB tax and substitution effects in other beverages or sugary foods [31]. Without a thorough investigation on substitution effects, the impacts of SSB tax might not be accurately estimated. Therefore, it is important to consider this perspective in the future. Moreover, the new excise SSB tax in Thailand is imposed on most all types of beverages in the market including fruit juice and diet carbonated drinks. Therefore, it is difficult to assess the exact cross-price elasticity from non SSB product to SSB per se. To tackle this challenge, a better comprehensive model is needed such as a system dynamic model.

Secondly, some key model parameters (pass on rate and own-price elasticity) were based on evidence from foreign studies rather than the empirical estimates from Thai population. This study assumed that the tax fully passed on to consumers. However, in reality, pass on rates might be under- or over-shifted. The price elasticity is a key parameter to estimate the power of purchasing/consumption when the price of a product is changed. The variations in price elasticity would yield different results. For example, a lower price elasticity applied in the study of Briggs et al. [15] and Veerman et al. [19] showed a smaller change in consumption and subsequently, a modest change in obesity prevalence when compared with other studies that applied a higher value of price elasticity. Primary research on applying these parameters is needed in the Thai context. To minimize this limitation, we conducted sensitivity analyses to assess the impact of SSB tax given varying degrees of pass on rate and price elasticity.

Thirdly, this study did not account for differential effects by income groups, due to lack of data on socio-economic status (SES). Previous studies have shown a significant association between income groups and the levels of SSB intake, in which people with lower income families consumed more SSB than those in high income families [45,46]. In Mexico, there was a greater decrease in SSB purchases among low SES households than those in higher ones, after two year of a 10% SSB tax implementation [47]. A study in Columbia showed that an SSB tax would reduce obesity prevalence at a greater degree in lower SES households [48]. Backholer et al. also reported that an SSB tax would have a greater impact on low-income groups than the more affluent groups [49]. Consequently, those in less affluent households were more likely to change their consumption behaviours and gain more health benefits than the richer [41,50]. Thus, estimating the impacts of an existing SSB tax on different income groups would provide a more fine-grained picture on consumption behaviours and health status across all populations.

Finally, this research focused only on BMI and obesity prevalence. Recently, many studies have assessed the impact of SSB tax on consumption [51] and dental health [52]. However, it would be more beneficial if the analysis could cover other health indicators related to sugar consumption, such as obesity, type-2 diabetes, and cardiovascular disease (CVD) [53,54]. Other studies have modelled the impacts of SSB tax on these health indicators. For example, a study in the Philippines highlighted that a 13% increase in SSB tax would avert 5,913 deaths from diabetes, 10,339 deaths from ischaemic heart disease and 7,950 deaths from stroke [53]. Moreover, dental problems among 269,375 persons would be reduced when implementing SSB tax policy in the UK [54]. Modelling the various health impacts as a cause from SSB consumption would provide a wide picture of the SSB tax impact on health.

The findings from this study suggested that the higher SSB tax rates might yield more favourable effect on consumption and health outcomes. A systematic review on the impact of SSB tax policy has confirmed that a lower tax rate (5%) has no effect on volume sales [55]. Another study, on the other hands, mentioned a significant decrease in purchasing a sugary drink with each increasing tax level up to 30% [56]. A small tax rate is insufficient to provide a meaningful impact on health outcomes and could take a long time before demonstrating a favourable effect [57], whereas a higher impact on health could be observed when applying a higher tax rate [52]. Evidence suggested that the tax rate should be up to 20% to reveal a better health outcome [10,58]. This evidence, to some extent, indicates the benefit of a higher tax rate on consumption and purchasing behaviours.

The main objective of SSB tax policy is to reduce sugar consumption, and therefore reducing diet-related risk factors for NCDs such as obesity and diabetes. Thailand has been following the global NCDs targets, which one of the targets is to halt the rise in diabetes and obesity [59,60]. The pathway effect of a tax policy would decrease purchasing and consumption through pricing mechanism [61]. A higher tax rate will accelerate beverage manufacturers to

lower sugar content in order to meet the threshold. Recent study showed a 9.6% decrease in sugar content in domestic SSB products, during a grace period of tax implementation, especially in carbonated drinks (18%) which mostly consumed among Thais [24]. Some beverage industries reformulated their product, as to avoid the tax. This would be beneficial for consumers as sugar intake could be minimized. Consequently, the benefit would transfer toward a national policy to meet the global NCDs targets on halting the rise of diabetes and obesity prevalence.

SSB taxes have been reported as one of the most cost-effectiveness policies to prevent obesity and NCDs [25]. Apart from health benefits, SSB taxes would generate revenue to the state, which can be utilized for a general fund or initiatives and projects related to health and/or NCD prevention to raise public awareness about sugar consumption as well as to trigger product reformulation to reduce sugar content [61,62]. However, SSB taxes policy alone would not provide fully potential effect on changing behaviour and health outcome [63]. People may avoid taxed products due to higher price somehow and would compensate with similar non-nutritive sweeteners or other caloric foods [64]. To comprehensively achieve the main aim of an SSB tax policy, multi-sectoral public policies to create a healthy environment, such as educational and health awareness campaigns [61] and food labelling [63], are indispensable to pave the way toward the goal.

Thailand already has an excise tax imposed on non-alcoholic beverages including SSB which has been in place for over a decade. New tax rates were recently promulgated for Thailand in 2017, however, the rates are not as high as recommended since the price of SSB increased by 11%. This study illustrated an estimated impact of a higher tax rate on a reduction in consumption, BMI and obesity prevalence. Therefore, Thailand could start learning from the impact of the current SSB tax while waiting for the window of opportunities to extend the tax rate for a better health impact.

## Conclusions

This study assessed the impacts of three different SSB tax rate scenarios on BMI and obesity prevalence in the Thai population through an economic-epidemiologic model. It is clear that the increase in SSB tax is estimated to reduce SSB consumption, body mass index and obesity prevalence in Thailand. The study findings present a similar trend of results with international literature. Future studies that delve into the impact of SSB on additional health outcomes apart from obesity prevalence with a consideration of different SES levels and diverse degrees of cross- price elasticity, are indispensable. This study may serve as a basis for more advanced research on the cost-effectiveness of SSB tax policy in the future.

## Supporting information

**S1 Table.** Baseline data of (a) daily SSB consumption in ml/day/person and (b) energy intake in kJ/person/day by sex and age groups among Thais.
(DOCX)

**S2 Table.** Baseline data of (a) weight (kg) and (b) body mass index (BMI) by sex and age groups among Thais.
(DOCX)

**S3 Table. The differences of sex and age groups and change in consumption, change in weight and change in BMI in three different tax scenarios.**
(DOCX)

## Acknowledgments

We would like to acknowledge the National Bureau of Agricultural Commodity and Food Standards (ACFS) and Institute of Nutrition, Mahidol University (INMU) for supporting the key input data on consumption and obesity prevalence. We also would like to thank Asst.Prof. Surasak Chaiyasong, Faculty of Pharmacy, Mahasarakam University, Thailand, and Jintana Jankhotkaew, from IHPP, for supporting methodological and data analysis advice.

## Author Contributions

**Conceptualization:** Payao Phonsuk, Suladda Ponguttha, Rapeepong Suphanchaimat, Nipa Rojroongwasinkul, Boyd Anthony Swinburn.

**Data curation:** Nipa Rojroongwasinkul.

**Methodology:** Payao Phonsuk, Vuthiphan Vongmongkol, Suladda Ponguttha, Rapeepong Suphanchaimat.

**Resources:** Nipa Rojroongwasinkul.

**Supervision:** Boyd Anthony Swinburn.

**Writing – original draft:** Payao Phonsuk.

**Writing – review & editing:** Vuthiphan Vongmongkol, Suladda Ponguttha, Rapeepong Suphanchaimat, Boyd Anthony Swinburn.

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
