## [Decision Letter · Decision Letter 0]

5 Jan 2021

PONE-D-20-23001

Impacts of a sugar sweetened beverage tax on body mass index and obesity in Thailand: a modelling study

PLOS ONE

Dear Dr.   Payao Phonsuk  

Thank you for submitting your manuscript to PLOS ONE. After careful consideration, we feel that it has merit but does not fully meet PLOS ONE’s publication criteria as it currently stands. Therefore, we invite you to submit a revised version of the manuscript that addresses the points raised during the review process.

I have now received 2 reviews of your MS which has been submitted to PloS One. Both reviewers have indicated scientific merit in the MS as submitted, but indicate the need for further ere visons before it is acceptable for publication. Both reviewers recommend proofreading, including an English language check, and reference check. In addition, both reviewers raise concerns about the analysis applied.  and the second reviewer about the underlying assumptions in relation to Thai taxation in relation to sugar tax.

If you are willing to revise the MS prior to publication, please address the following issues in particular, as well as all recomm3endations made by the reviewers.

1. The need to proof-read the paper thoroughly, including an English language check

2.  Application of MANOVA to check for significant age effects (Reviewer 1)

3/. Details about the Thai sugar tax regime (Reviewer 2)

4. Provide a more detailed discussion regarding the implications of your findings, in particular in relation to practical policy implications.

We look forward to receiving your revised manuscript.

Kind regards,

Lynn Jayne Frewer, MSc PhD

Academic Editor

PLOS ONE

Additional Editor Comments:

I have now received 2 reviews of your MS which has been submitted to PloS One. Both reviewers have indicated scientific merit in the MS as submitted, but indicate the need for further ere visons before it is acceptable for publication. Both reviewers recommend proofreading, including an English language check, and reference check. In addition, both reviewers raise concerns about the analysis applied. and the second reviewer about the underlying assumptions in relation to Thai taxation in relation to sugar tax.

If you are willing to revise the MS prior to publication, please address the following issues in particular, as well as all recomm3endations made by the reviewers.

1. The need to proof-read the paper thoroughly, including an English language check

2. Application of MANOVA to check for significant age effects (Reviewer 1)

3/. Details about the Thai sugar tax regime (Reviewer 2)

4. Provide a more detailed discussion regarding the implications of your findings, in particular in relation to practical policy implications.

3. Please ensure that you refer to Figures 1 and 2 in your text as, if accepted, production will need this reference to link the reader to the figure.

Reviewers' comments:

Reviewer's Responses to Questions

**Comments to the Author**

1. Is the manuscript technically sound, and do the data support the conclusions?

Reviewer #1: Partly

Reviewer #2: Partly

2. Has the statistical analysis been performed appropriately and rigorously? 

Reviewer #1: Yes

Reviewer #2: Yes

3. Have the authors made all data underlying the findings in their manuscript fully available?

Reviewer #1: Yes

Reviewer #2: No

4. Is the manuscript presented in an intelligible fashion and written in standard English?

Reviewer #1: Yes

Reviewer #2: No

5. Review Comments to the Author

Reviewer #1: In is an interesting article that could help in policies to reduce the prevalence of obesity. Overall the manuscript is well written but need several improvements:

1. Results.

- Comparison were made between men and women as well as different age groups for all data but there have been no mention whether the differences were significant or not. There was a mention about differences in energy intake across age groups but no mention about the effect of age groups for the other data/variables. Does this mean for the other data/variables, there were no effect of ages?. If the model do not provide significant tests, it is suggested to carry out MANOVA or at least ANOVA to test whether the differences are significant.

- Sensitivity analysis in page 14, line 229-237 should be combined with the similar topics in page 10.

- Add a note below the Table for shortforms that has not been mentioned before. Eg. CI

2. Discussions.

- It is suggested to add implications of the findings.

- Line 248, list the references for the other studies.

3. Need minor proof reading of English.

Reviewer #2: This is yet another modeling study evaluating the impact of SSB tax on obesity. There have been similar modeling studies published in the past. The advantage of this study is that it provides the exercise from a LMIC--Thailand.

However, I have some major and minor concerns as followed.

Major

1. Regarding the actual Thai tax regime, what is the total(calculate) tax rate in each level? These actual tax rate should be used as the based model to explain the real situation of your country. The 20% tax rate as WHO recommended can be used as a scenario to show the discrepancy between the real situation and the recommendation. Moreover, the recent study in Thailand, Markchang et al(2019) showed the evidences of actual SSB price changes and SSB sugar content change. This evidences need to be consider in the model. This might explain the real pass-on rate and the calories from sugar content in SSB as well.

All other scenario which the authors used sensitivity analysis are acceptable to show the uncertainty and compliment the model.

2. The SSB tax has already been introduced in Thailand since 2017. This is possible already enough time to change consumption rates. If you can find the recent consumption data after the tax implementation, this might be reasonable and more reliable than assuming on price elasticity.

Minor

1. Please check the accuracy of all your references and revise, i.e. ref#7 and #8 in line 51-54.

2. The manuscript is in need of proofreading and correct some typo and grammatical error, i.e. line 79, line 225. Please also make it's consistent between the word "obesity" and "overweight" which one you really mean to.

6. PLOS authors have the option to publish the peer review history of their article (what does this mean?). If published, this will include your full peer review and any attached files.

Reviewer #1: No

Reviewer #2: No

---

## [Author Response · Author response to Decision Letter 0]

22 Mar 2021

Response: The authors have checked the criteria requirements and have improved the draft accordingly.

Response: Concerning the availability of data used in this study, we have stated that data are available upon request. To elaborate on this, the data is owned by the National Bureau of Agricultural Commodity and Food Standards (ACFS), Thailand. We, thus, have officially asked for their permission to use the data in our study. However, we can provide the minimal data set to replicate study findings as in the supporting information (S1 Table and S2 Table). Furthermore, the DOIs for this data set have been created.

3. Please ensure that you refer to Figures 1 and 2 in your text as, if accepted, production will need this reference to link the reader to the figure.

Response: The authors have mentioned Fig 1 and Fig 2 also the supporting information in the draft.

Reviewer #1: In is an interesting article that could help in policies to reduce the prevalence of obesity. Overall, the manuscript is well written but need several improvements:

1. Results.

- Comparison were made between men and women as well as different age groups for all data but there have been no mention whether the differences were significant or not. There was a mention about differences in energy intake across age groups but no mention about the effect of age groups for the other data/variables. Does this mean for the other data/variables, there were no effect of ages?. If the model do not provide significant tests, it is suggested to carry out MANOVA or at least ANOVA to test whether the differences are significant.

- Sensitivity analysis in page 14, line 229-237 should be combined with the similar topics in page 10.

- Add a note below the Table for shortforms that has not been mentioned before. Eg. CI

Response: The MANOVA analysis was employed to test the differences of age groups and also gender on the dependent variables including change in SSB consumption, change in weight and change in BMI. The results showed a statistically difference between these variables, and has been added to the results. The accordant of sensitivity analysis topic was amended. The definition of abbreviations used in the model and entire manuscript were included. 

2. Discussions.

- It is suggested to add implications of the findings.

- Line 248, list the references for the other studies.

Response: The implication of the findings was discussed more on the importance of tax rates and the impacts of consumption, purchasing behavior and health outcome. Outcomes from a systematic review were added. We also considered the policy situation in the country and found that only SSB tax policy may not be fully working and have stated the importance of multi-sectoral approaches. We also addressed the reference lists of the studies we mentioned in the discussion part. 

3. Need minor proof reading of English.

Response: This revised manuscript was proofread by an English professional user who has experienced in teaching and editing English documents.

Reviewer #2: This is yet another modeling study evaluating the impact of SSB tax on obesity. There have been similar modeling studies published in the past. The advantage of this study is that it provides the exercise from a LMIC--Thailand.

However, I have some major and minor concerns as followed.

Major

1. Regarding the actual Thai tax regime, what is the total(calculate) tax rate in each level? These actual tax rate should be used as the based model to explain the real situation of your country. The 20% tax rate as WHO recommended can be used as a scenario to show the discrepancy between the real situation and the recommendation. Moreover, the recent study in Thailand, Markchang et al(2019) showed the evidences of actual SSB price changes and SSB sugar content change. This evidences need to be consider in the model. This might explain the real pass-on rate and the calories from sugar content in SSB as well.

All other scenario which the authors used sensitivity analysis are acceptable to show the uncertainty and compliment the model.

Response: This study used a percentage increase in SSB price (11%) to reflect the current situation of SSB tax policy in Thailand. The structure of SSB tax is quite complex and the tax rates are different among SSB products. Also, there was no data available for the price elasticity and pass-on rate in Thailand. We, therefore, run a model on a 11% price increase as the result of an imposed tax. This is a main limitation of our model and we suggest the future research to cover this issue. 

2. The SSB tax has already been introduced in Thailand since 2017. This is possible already enough time to change consumption rates. If you can find the recent consumption data after the tax implementation, this might be reasonable and more reliable than assuming on price elasticity.

Response: The study on the change in SSB consumption after tax was just published in December, 2020. The results showed a 2.5% decrease in SSB consumption (Phulkerd et. Al, 2020). This study has taken this point in the discussion part. However, to assess the price elasticity, some data is needed such as the recent price of SSB. However, the availability of SSB price is in 2017, where it is the first wave of tax implementation. Presently, we are in the second wave of SSB tax policy and there are no studies to estimate change in price of SSB yet.

1. Please check the accuracy of all your references and revise, i.e. ref#7 and #8 in line 51-54.

Response: We have adjusted the list of references according to the comment.

2. The manuscript is in need of proofreading and correct some typo and grammatical error, i.e. line 79, line 225. Please also make it's consistent between the word "obesity" and "overweight" which one you really mean to.

Response: We have adjusted the typo and grammatical error, and made a consistency of words according to the comment.

---

## [Editor Report · Decision Letter 1]

15 Apr 2021

Impacts of a sugar sweetened beverage tax on body mass index and obesity in Thailand: a modelling study

PONE-D-20-23001R1

Dear Dr.Phonsuk  

We’re pleased to inform you that your manuscript has been judged scientifically suitable for publication and will be formally accepted for publication once it meets all outstanding technical requirements.

Kind regards,

Lynn Jayne Frewer, MSc PhD

Academic Editor

PLOS ONE

---

## [Editor Report · Acceptance letter]

20 Apr 2021

PONE-D-20-23001R1 

Impacts of a sugar sweetened beverage tax on body mass index and obesity in Thailand: a modelling study 

Dear Dr. Phonsuk:

I'm pleased to inform you that your manuscript has been deemed suitable for publication in PLOS ONE. Congratulations! Your manuscript is now with our production department. 

Kind regards, 

on behalf of

Dr. Lynn Jayne Frewer 

Academic Editor

PLOS ONE